# TopoFormer: Topology-aware Transformer for Reactive Motion Prediction in Close Interactions

## Abstract

With high-quality motion datasets more accessible, data-driven modelling of close interactions between two or more people has attracted more research interest in recent years. Such models can be used to understand the intent of the people by predicting the reactive motion when they are closely interacting with each other. However, failure in synchronising the motions between people as well as implausible motions such as interpenetrations of body parts can still be found in State-of-the-Art (SOTA) interaction prediction approaches. We argue that commonly used motion representations in Euclidean space, such as joint positions and joint angles in previous approaches do not capture the spatial relations between the body parts effectively. In this paper, we propose a new Transformer, called 'TopoFormer', for predicting the reactive motion of one of the characters in a Two-person close interaction by giving the motion of the other character and the interaction class label as input. TopoFormer consists of a Topology-Aware Spatio-Temporal Embedding and Spatial Relation-aware Multi-Headed Self Attention (SR-MSA) to facilitate the learning of the latent representation of close interactions. By representing the body parts using a set of articulated chains instead of the commonly used graph-based structure in recent works, the spatial relations can be more effectively represented using a topology-based representation, Gauss Linking Integral (GLI). Experimental results highlight the effectiveness of our proposed method as we achieved SOTA performance in Aligned Mean Error (AME) and a newly proposed metric Average Interpenetration per Frame (AIF) across different datasets and qualitatively more synchronised and plausible interactions.

## 1 Introduction

Understanding the intent of people who are closely interacting with each other, such as having a lot of body contact and tangling of the body parts/limbs, can be used in a wide range of downstream applications including group behavior understanding, human-robot interactions, crowd simulations, etc. While there have been huge advances in modelling single human motion in recent years, handling close interactions with two or more characters is still a challenging task. Even with a high-quality motion dataset (Guo et al., 2022; Fieraru et al., 2020) for training a state-of-the-art deep neural network for interaction prediction Chopin et al. (2023); Guo et al. (2022); Men et al. (2022); Goel et al. (2022); Peng et al. (2023); Tanke et al. (2023); Xu et al. (2023) , recent work in SOTA reactive motion prediction method (Chopin et al., 2023) still generates asynchronous interactions and implausible motions such as interpenetration of body parts.

The core difficulty of representing the interaction semantics in skeletal motions lies in its raw representation. Existing methods employ joint position, angles, velocities, etc. either/both in the Euclidean space or/and a skeleton-based topological graph. These representations do not capture the semantics at the posture level in interactions. Taking hugging as an example. The semantics of hugging dictate one person's arms surrounding the other person's torso. The geometric details such as the wrist positions are insignificant as long as the 'surrounding' exists, e.g. over the shoulder or around the waist. This suggests a *topological* feature at play instead of a geometric one. Furthermore, most topological features (betti numbers, geometric genus) are discrete, and unfriendly

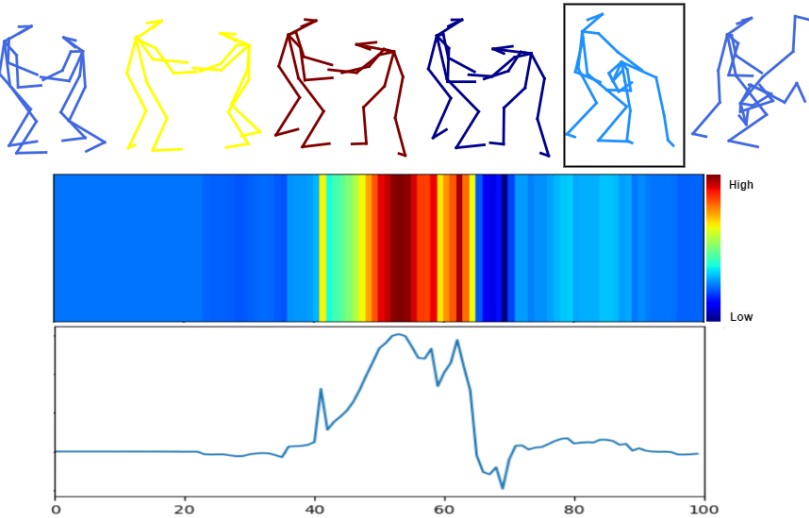

Figure 1: Effective Receptive Field (ERF) (Luo et al., 2016) given the frame of interest (in black box) of the cartwheel motion. Red regions correspond to high dependency. **Top**: Highlights which of the motions importance as per our frame of interest. **Middle**: The heatmap of each frame as per our proximal bias of Relative Positional Encoding. **Bottom**: The line graph of the ERF as per the frames of interaction. Our TopoFormer is able to learn the topological and proximal relationship across the spatio-temporal domain of the interaction.

to deep learning. As a result, a topological feature with continuous parameterization is needed to describe the interaction semantics.

To tackle this problem, topology-based motion representations Ho & Komura (2009); Ho et al. (2013) have been proposed for synthesising close interactions. Specifically, Gauss Linking Integral (GLI) was used in the aforementioned approaches for representing body parts in proximity. GLI is a signed scalar value which can be computed from two 3D curves. Imagine the serial chains illustrated in Figure 3 (right) represent a pair of entangled limbs of the character(s), switching between the left and right configurations will easily result in interpenetrations which is indicated by a significant change in the GLI values (i.e. sign change). By minimizing the changes of the pairwise GLIs over consecutive frames, the interpenetration of body parts can be avoided.

In this work, we propose a new Transformer, called *TopoFormer*, for predicting the reactive motion of one of the characters in a two-person close interaction by giving the motion of the other character and the interaction class label as input. TopoFormer consists of a Topology-Aware Spatio-Temporal (TST) embedding and Spatial Relation-aware Multi-Headed Self Attention (SR-MSA) to facilitate the learning of the latent representation of close interactions. By representing the body parts using a set of articulated chains instead of the commonly used graph-based structure in recent works, the spatial relations can be more effectively represented using GLI.

To demonstrate the effectiveness of our method, we compared our method with SOTA approaches on the reactive motion prediction task on the ExPI (Guo et al., 2022) and CHI3D (Fieraru et al., 2020) datasets with different settings. Our method achieved SOTA performance in Aligned Mean Error (AME) and a newly proposed metric Average Interpenetration per Frame (AIF), which produced qualitatively more synchronised and plausible interactions.

## 2 RELATED WORK

Learning from motion data became the mainstream for close interaction generation in recent years as two-person interaction datasets (Shen et al., 2020; Guo et al., 2022; Fieraru et al., 2020) are more accessible. Existing methods are mainly focusing on **interaction prediction** (i.e. given the initial motions of two people, the future motions will be predicted) and **reactive motion prediction** (i.e. given the motion of one person, the motion of the other person will be predicted).

**Interaction Prediction**   While motion prediction for a single person is an active research area, predicting the motions of multi-person simultaneously is relatively new to the community. Rahman et al. (2023) applied several best practices in single-person motion prediction to the multi-person prediction problem. Kundu et al. (2020) proposed using two recurrent neural networks, namely Cross-Conditioned Recurrent Networks, to predict the future motion of another character according to the initial motion input for synthesizing two-person interactions. The 2-stream cross-prediction approach is also proposed in (Guo et al., 2022) with the cross-interaction attention (XIA) module. For interactions with relatively fewer body contacts such as those in social interactions, Social Diffusion (Tanke et al., 2023) was used for modelling the distribution of multi-person interactions using a diffusion-based framework. Xu et al. (2023) proposed DuMMF to model social interactions using a dual-level model, with the local level for individual motion while the global level for social interaction by taking into account the correlations between the motions of the people in the scene. TBIFormer (Peng et al., 2023) forecasts the 3D pose sequences of multi-person using a transformer-based architecture. However, the human skeletal structure was modelled in an abstract way by performing an average pooling on each of the 5 body parts. In contrast, our proposed method models all body joints based on a topology-aware spatiotemporal embedding. This can more effectively model close interactions such as hugging and dancing.

**Reactive Motion Prediction**   A GAN-based model with dual discriminators has been proposed in (Men et al., 2022) to generate the follower's motion. By having one discriminator for real/fake and the other one for interaction classification, the quality of the synthesized motion can be improved. Goel et al. (2022) proposed a conditional hierarchical GAN which models each character at full-body level and limb-level. The follower's motion is then synthesized according to the leader's motion and the interaction class label. InterFormer (Chopin et al., 2023) is the first transformer-based method for reactive motion prediction which models the poses using a graph representation.

**GLI for Close Interaction Generation**   GLI can be used for representing the topology between articulated objects such as human body parts. Ho & Komura (2007) proposed using PD-control to maintain the GLI values between body parts to preserve entangled limbs in wrestling motions. The topology-based pose representation can also be used with local Principal Component Analysis (lPCA) to construct a latent space for synthesizing close character-object interactions (Ho et al., 2013). Topology Coordinates (Ho & Komura, 2009) was proposed to further improve the controllability of the synthesis of tangling motions in human-human and human-object interactions. By modifying the distribution of the GLI values in the *writhe matrix*, the 3D configuration of the interacting body parts can be controlled intuitively in complex tasks such as tangling and untangling.

## 3   METHODOLOGY

An overview of the network components is illustrated in Figure 2. The spatial and topological features (Section 3.1) are fed into the Topology-Aware Spatio-Temporal (TST) block (Section 3.2) for motion embedding. The proposed Spatial Relation-aware Relative Position Encoding (srRPE, Section 3.3) and Spatial Relation-aware Multi-head Self-Attention (SR-MSA, Section 3.4) further learn the importance of the interacting body parts in proximity. Finally, the predicted reactive motion will be reconstructed into 3D skeletal motion using the Transformer Decoder (Section 3.5).

### 3.1   SPATIAL AND TOPOLOGICAL FEATURES

Given the 3D positions of the joints in each frame, the spatial information is represented as a 1-D vector. The skeletal structure we used consists of 18 joints (for ExPI, 19 for CHI3D), and this results in a 1-D vector with 54 elements.

Inspired by the success of using GLI as soft constraints in an optimization-based approach Ho & Komura (2009) for generating collision-free close interactions, we propose encoding the pairwise GLIs of closely interacting body parts to prevent implausible motions such as interpenetrations.

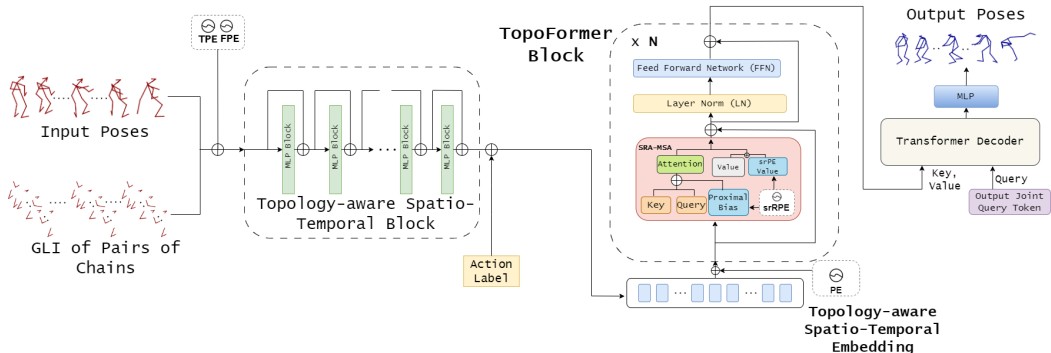

Figure 2: Our overview for TopoFormer. On the left, our Topology-aware Spatio-Temporal Block takes the input interaction poses and Gauss Linking Integrals for each pair of chains alongside Temporal Position Encoding (TPE) and Frame Position Encoding (FPE) to get the Topology-aware Spatio-Temporal (TST) Embedding. On the right, the TST Embedding is concatenated with a standard transformer positional encoding to our TopoFormer Blocks. Our Spacial Relation-aware Relative Position Encoding is incorporated in each SR-MSA block. Our TopoFormer encodings are used as Key and Values for a standard Transformer Decoder to get the output poses.

**Gauss Linking Integral (GLI)**    Given two directed curves $\gamma_1$ and $\gamma_2$, the Gauss Linking Integral (GLI) (Pohl, 1968) can be calculated by Eq. 1 by integrating along the two curves:

$$G(\gamma_1, \gamma_2) = \frac{1}{4\pi} \int_{\gamma_1} \int_{\gamma_2} \frac{d\gamma_1 \times d\gamma_2 \cdot (\gamma_1 - \gamma_2)}{\|\gamma_1 - \gamma_2\|^3} \tag{1}$$

where $\times$ and $\cdot$ are cross-product and dot-product operators, respectively. GLI is a signed scaler value which computes the average number of crossings from all viewing directions of the two curves, such as Figure 3 (right). Please refer to the Supplementary Material (SM) for the discretization of Eq. 1 and the analytical solution for GLI calculation.

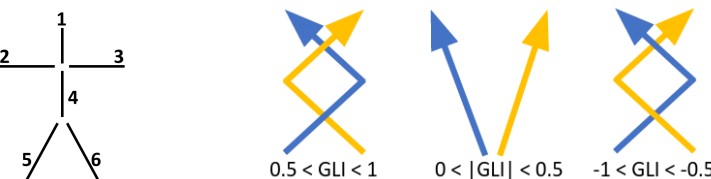

Figure 3: Representing a skeletal pose by 6 serial chains (left). Examples of GLI values computed from different configurations of a pair of serial chains representing the body parts (right).

**Representing Skeletal Pose using GLI**    We divide the ground truth character sequence into 6 serial chains in Figure 3 $\{S_{i,t}^A\}_{i=1}^6$ for each frame $t$. Since GLI can be used as a topological representation of two curves, the poses have to be represented by a set of serial chains. Representing each limb and torso using a serial chain (Figure 3 left) is an intuitive choice. With 6 serials chains in each pose, 15 pairwise GLI will be calculated from each frame in the input motion.

## 3.2    TOPOLOGY-AWARE SPATIO-TEMPORAL (TST) BLOCK

The extracted spatial and topological features are concatenated as a 1-d vector with $54 + 15 = 71$ elements (for the ExPI dataset) and passed into an encoder to further extract the latent representation. Inspired by the superior performance obtained using basic MLP networks with positional encoding in NeRF (Mildenhall et al., 2020) and a recent study on human motion prediction (Guo et al., 2023), we propose to use an MLP with 8 hidden layers as the TST architecture. Each MLP layer is followed by a ReLu activation (Agarap, 2018). Furthermore, we facilitate the usage of two

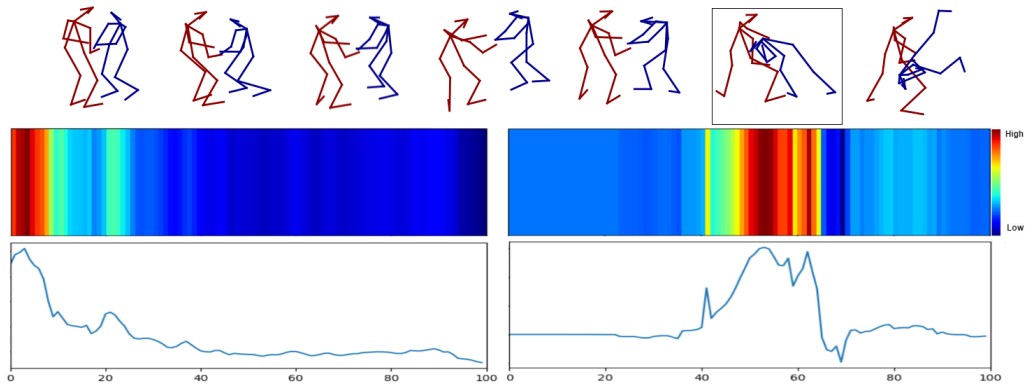

Figure 4: Effective Receptive Field (Luo et al., 2016) of the proximal bias given the frame of interest (in black box) of the cartwheel motion of the last head of the last transformer block. Red regions correspond to high dependency. At the **(top)** is the ground truth cartwheel motion. **(left)** is with MLP positional encoding and **(right)** is with our srRPE.

positional encoding (Mildenhall et al., 2020), Temporal Positional Encoding (TPE) and Frame Positional Encoding (FPE). The TPE maps the temporal coordinates from $1 : T$. The FPE maps the spatial and topological frame features from $1 : D$. The output of the TST Block is concatenated with the action label of the class interaction to form the TST embedding.

### 3.3 SPATIAL RELATION-AWARE RELATIVE POSITION ENCODING (SRRPE)

In close interactions, body segments in close proximity are usually more important as they provide the context of the interactions. For example, the hands are in contact in the 'high-five' action. In addition, those body segments have to be coordinated carefully when editing or synthesizing the motions to avoid generating infeasible motions with interpenetrations. Here, we propose a novel interaction-aware relative position encoding based on the body part-based structure of the two characters. Given the 15 unique pairs of chains as explained in Section 3.1, we compute the *minimum* distance between the joint locations on the 2 chains in each pair to get **srRPE**$\epsilon\mathbb{R}^{Tx15}$. Finally, the minimum joint-joint distance is mapped to a set of integer relative position encoding using a piecewise index function (Wu et al., 2021):

$$g(x) = \begin{cases} |x|, & |x| \leq \alpha \\ sign(x) \times min(\beta, [\alpha + \frac{\ln{(|x|/\alpha)}}{\ln{(\gamma/\alpha)}}(\beta - \alpha)]), & |x| > \alpha \end{cases} \qquad (2)$$

. By specifying the values for $\alpha$, $\beta$ and $\gamma$, we can control the piecewise point, output range and curvature of the logarithmic parts, respectively.

### 3.4 SPATIAL RELATION-AWARE MULTI-HEAD SELF-ATTENTION (SR-MSA)

In each encoder block of the TopoFormer, the Spatial Relation-aware Multi-Head Self-Attention (SR-MSA) is constructed to incorporate the proximity of the interaction's input motion chains. Each SR-MSA block has a common srRPE which is mapped by three learnable lookup tables, Key, Query and Value. **srRPEk**, **srRPEq**, **srRPEv**$\epsilon\mathbb{R}^{TxN_h}$. For a multi-head self-attention of heads $m$, if each head has a dimension of $N_h$ then $N_d = mxN_h$ is the dimension of the input feature $x\epsilon R^{Tx(mxN_h)}$ to the SR-MSA block. Specifically, it is formulated as

$$\mathbf{q} = W_q x, \qquad \mathbf{k} = W_k x, \qquad \mathbf{v} = W_v x,$$

$$\mathbf{proximalbias}_{i,j,h} = \mathbf{q}_{i,h} \cdot \mathbf{srRPEq}_{i,h} + \mathbf{k}_{j,h} \cdot \mathbf{srRPEk}_{i,h},$$

$$\mathbf{attnsrRPE}_{i,j,h} = \mathbf{q}_{i,h} \cdot \mathbf{k}_{j,h} + \mathbf{proximalbias}_{i,j,h},$$

$$\hat{\mathbf{attnsrRPE}}_{i,.,h} = softmax(\frac{\mathbf{attnsrRPE}_{i,.,h}}{\sqrt{Nh}}),$$

$$y = \sum_{j=1}^{L} \hat{\mathbf{attnsrRPE}}_{i,j,h}(v_{j,h} + srRPEv_{i,j,h}).$$

Our proximal bias provides adaptive weighted information about the proximal nature of the interaction frames. This has more semantic information as compared to an MLP-positional encoding which maps the Euclidean x,y,z of the pose via MLP to a proximal bias. The proximal bias is visualized in Figure 4. We can see that the starting of the tumbling motion of the cartwheel is captured by our sr-RPE whereas the MLP-positional encoding has a decaying proximalbias dependency. Furthermore, experimental results in Tables 6 and 4 show the superiority of the srRPE. We use LayerNorm (Ba et al., 2016) after each SR-MSA block.

### 3.5 TRANSFORMER DECODER

A standard Transformer decoder (Vaswani et al., 2017) is used in our proposed network for generating the output (i.e. 3D skeletal pose) through modelling the relationship between the queries and keys. Specifically, the Key and Value token is the output of our proposed TopoFormer Block. The Query token is taken as the output of the reactive poses mapped with an MLP.

### 3.6 LOSS FUNCTION

As the ultimate goal is to predict the follower's motion based on the leader's motion as input, we propose to use a reconstruction loss to measure the quality of the predicted follower's motion. Specifically, we used the Mean Per Joint Position Error (MPJPE):

$$L = \frac{1}{J * N} \sum_{i=N+1}^{T+N} \sum_{j=1}^{J} ||\hat{\boldsymbol{y}}_{i,j} - \boldsymbol{y}_{i,j}||^2 \tag{3}$$

where $J$ is the total number of joints, and $\hat{y}_{i,j}$ and $y_{i,j}$ are the 3D positions of the $j$-th joint at $i$-th frame in the ground-truth and predicted motions, respectively.

## 4 EXPERIMENTAL RESULTS

### 4.1 DATASETS

In this section, the details of the 2 publicly available datasets used in our experiments will be presented. To facilitate the learning process, We normalized all motions by removing the global translation and rotation about the vertical axis such that the first person is forward-facing and located at the origin.

**Extreme Pose Interaction (ExPI) Dataset** (Guo et al., 2022) is a recently published dataset which contains 115 high-quality close two-person interaction sequences from 7 classes, such as the Cartwheel interaction, captured using an optical MOCAP system. Each interaction is provided as the 3D joint position sequences of the two characters. There are two protocols used in our experiments: 1) Cross-Subject protocol (known as 'common action split') used for the interaction prediction task as in Guo et al. (2022), 2) Cross-Trial protocol by gathering the 2 couples who performed all common interaction classes (as in the CS protocol) in the dataset, with a 7:3 training/testing ratio.

**Close Human Interactions 3D (CHI3D) Dataset** (Fieraru et al., 2020) is a high-quality 3D motion dataset which contains 631 two-person interaction sequences captured from 6 human subjects grouped up in 3 pairs. The dataset includes various types of interactions, such as Grabbing, Handshaking, Hitting, Holding Hands, Hugging, Kicking, Posing, and Pushing. We grouped the 3 subject pairs interactions and divided them using a 7:3 training/testing ratio with a Cross-Trial protocol.

For evaluation purposes, we further scaled the motions spatially such that the AME to be computed is comparable to those obtained from the ExPI dataset. We assume the first subject in each pair is 170cm tall (head to toe) to compute the scaling factor. The fingers, thumbs and toes were removed to reduce the total number of joints from 25 to 19 for each pose. Also, the motion sequences were downsampled from 50 fps to 25 fps to reduce the temporal resolution.

## 4.2 IMPLEMENTATION DETAILS

The code base is built upon Pytorch library. The experiments are conducted using Nvidia RTX 2080 Ti Graphics card, Adam Optimizer (Kingma & Ba, 2014) with a learning rate of $1 \times 10^{-4}$, batch size of 6, and 50 epochs for all datasets. Each SR-MSA block has $m = 4$ heads and $N_D = 256$. For srRPE, $\alpha = 0.001$, $\beta = 90$ and $\gamma = 16000$ is set for all datasets. We set $L = 7$ for our TPE and FPE (Mildenhall et al., 2020). For TST Block, there are 7 MLP layers with hidden dimension 256 and 8th MLP layer with hidden dimension 50. There are 6 TopoFormer blocks for our encoder.

## 4.3 EVALUATION METRICS

The following metrics were used for evaluating the performance of reactive motion prediction:

**Aligned Mean Error (AME)** (Guo et al., 2022) measures the Mean Per Joint Position Error (MPJPE) between the ground truth $\hat{G}$ and the predicted $\hat{P}$ reactive motions: $\text{AME}(P, G) = \text{MPJPE}(T_A(\hat{P}, \hat{G}), \hat{G})$, where $T_A$ is the best rigid alignment between the normalized motions $\hat{P}$ and $\hat{G}$. Such an approach ignores the position bias while focusing on the error in the predicted pose.

**Average Interpenetration per Frame (AIF)** measures the absolute difference in GLI of two articulated chains for the ground truth (GT) and predicted (p) motion over two consecutive frames exceeds a threshold and indicates if an interpenetration has taken place:

$$\text{AIF}(A, B) = \frac{1}{T} \sum_{t}^{T} \sum_{i}^{6} \sum_{j,}^{6} f(A_i, B_j, t)$$

$$f(a, b, t) = \begin{cases} 1, & |G(a_{t+1}^p, b_{t+1}^p) - G(a_t^p, b_t^p)| > 0.5 \text{ and } |G(a_{t+1}^{GT}, b_{t+1}^{GT}) - G(a_t^{GT}, b_t^{GT})| \le 0.5 \\ 0, & otherwise \end{cases}$$

. where $A$ and $B$ contain the 6 serial chains for representing the skeletal pose of each person.

## 4.4 QUANTITATIVE EVALUATION

We compare our proposed method against the recent deep learning based approaches, including Men et al. (2022); Goel et al. (2022) and state-of-the-art (SOTA) InterFormer (Chopin et al., 2023) in the literature on the reactive motion prediction task.

For the ExPI dataset, the results of the Cross-Trial (CT) protocol are presented in Table 1 (leftmost values in each column). The results show that our proposed method achieved the lowest average AME (on all interaction classes) across all 10 different prediction duration (from 0.2s to 4s), which highlights the proposed method generates motions which are more similar to the ground truth.

Compared with SOTA InterFormer (Chopin et al., 2023), it can be seen that our method achieves a much lower AME, which is around 21% to 48% lower, in all prediction duration. Among all the methods, there is a general trend in increasing AME when the prediction duration increases. This aligns with the assumption that the prediction task becomes more challenging when the prediction duration increases. Nevertheless, our method shows a more significant advantage over InterFormer with a 48.39% and 46.03% lower AME under the prediction duration of 3.6s and 4.0s, respectively.

We further challenge our method by using the Cross-Subject (CS) protocol introduced in Guo et al. (2022) for the interaction prediction task. By using the CS protocol, the skeletal structures (e.g. bone lengths) in the testing set are unseen during training which brings a huge challenge to the reactive motion prediction task. On the other hand, it is less of a problem for the interaction prediction tasks as the initial observations (i.e. pose sequences) of both people are available during inference.

From Table 1 (rightmost values in each column), it can be seen that the AME obtained under the CS protocol is higher than those from the CT protocol. The overall trend is similar to the CT

Table 1: The AME of the predicted reactive motion under different prediction durations in the ExPI dataset. xx/xx are the results under the CT/CS protocols.

|  | AME (averaged across actions) (↓) - CT/CS | | | | | | | | | |
| --- | --- | --- | --- | --- | --- | --- | --- | --- | --- | --- |
| Time (sec) | 0.2 | 0.6 | 1.0 | 1.4 | 2.0 | 2.4 | 2.8 | 3.2 | 3.6 | 4.0 |
| Men et al. (2022) | 80/90 | 80/94 | 81/98 | 82/104 | 84/115 | 86/115 | 87/116 | 88/117 | 90/119 | 91/122 |
| Goel et al. (2022) | 71/82 | 72/85 | 72/87 | 74/90 | 76/93 | 78/108 | 80/109 | 80/110 | 82/113 | 85/115 |
| Chopin et al. (2023) | 32/61 | 36/64 | 40/70 | 45/74 | 47/77 | 57/92 | 57/96 | 58/97 | 62/96 | 63/94 |
| Ours | **25/58** | **25/60** | **26/62** | **28/63** | **28/65** | **30/70** | **30/71** | **31/74** | **32/76** | **34/78** |

results, with the AME increasing with the prediction duration, as well as our proposed method outperforms all the other methods. Note that the difference in the AME between our method and SOTA InterFormer (Chopin et al., 2023) is getting smaller under the CS protocol. However, it is worth noting that InterFormer (Chopin et al., 2023) requires the first frame of the reacting person as input to take advantage of facilitating the prediction using the 'known' skeletal structure.

To evaluate the robustness of our method, we further conducted the experiments on CHI3D (Fieraru et al., 2020) and the results are shown in Table 2. The maximum prediction duration is shorter because some of the motion sequences in the dataset only have 3.0s. Our method outperforms all other methods across all prediction durations, with around 5% to 26% lower in AME than InterFormer.

Table 2: The AME of the predicted reactive motion under different prediction durations in the CHI3D dataset.

|  | AME (averaged across actions) (↓) | | | | | | | |
| --- | --- | --- | --- | --- | --- | --- | --- | --- |
| Time (sec) | 0.2 | 0.6 | 1.0 | 1.4 | 2.0 | 2.4 | 2.8 | 3.0 |
| Men et al. (2022) | 81 | 83 | 86 | 87 | 89 | 90 | 93 | 94 |
| Goel et al. (2022) | 73 | 76 | 77 | 80 | 82 | 84 | 85 | 88 |
| InterFormer (Chopin et al., 2023) | 62 | 66 | 67 | 68 | 68 | 69 | 71 | 74 |
| Ours | **46** | **47** | **50** | **52** | **54** | **55** | **59** | **60** |

Another important property for introducing the TST block is to avoid interpenetration of body parts in the predicted motions. Table 3 presents the AIF in the ExPI and CHI3D datasets. In general, the ExPI dataset is more challenging since there are more body contacts in the interactions which more easily results in interpenetrations. Also, ExPI (CS) is more challenging than ExPI (CT) as discussed in the previous sections. Nevertheless, our method outperforms all other methods by achieving a lower AIF consistently across all datasets and protocols.

Table 3: The averaged number of interpenetrations per frame in the generated motions in the ExPI and CHI3D datasets.

| Method | AIF (↓) | | |
| --- | --- | --- | --- |
|  | ExPI (CT) | ExPI (CS) | CHI3D |
| Men et al. (2022) | 0.04015 | 0.05384 | 0.00409 |
| Goel et al. (2022) | 0.03215 | 0.04847 | 0.00388 |
| InterFormer (Chopin et al., 2023) | 0.02140 | 0.02992 | 0.00329 |
| Ours | **0.01521** | **0.02643** | **0.00250** |

## 4.5 ABLATION STUDY

Here, we present the results to highlight the importance of different components in our method.

**Topology-aware Spatio-Temporal Block**  The effectiveness of including the TST embedding is evaluated in an ablation study on the ExPI dataset using the CT protocol (Table 4). The results show that both the TST embedding and srRPE are needed to achieve the best performance in our proposed method. It can also be seen that srRPE has a more positive impact on the quality of the prediction motion. In addition to AME, AIF is also another important metric for evaluating close interactions. The AIF results of our ablation studies are presented in Table 5. In particular, the AIF increases more significantly when the TST block is not included in the network. This highlights the

TST block facilitates the learning of a topology-aware latent space which can effectively reduce the generation of implausible motions with interpentrations of the body parts.

Table 4: Ablation Study on Topology Aware Spatio-Temporal Embedding and srRPE. The ExPI dataset and Cross-Trial (CT) protocol were used.

| | AME (averaged across actions) (↓) | | | | | | | | | |
|---|---|---|---|---|---|---|---|---|---|---|
| Time (sec) | 0.2 | 0.6 | 1.0 | 1.4 | 2.0 | 2.4 | 2.8 | 3.2 | 3.6 | 4.0 |
| No TST Embedding | 29 | 29 | 29 | 30 | 31 | 32 | 34 | 35 | 35 | 36 |
| No srRPE | 33 | 33 | 33 | 35 | 35 | 36 | 37 | 38 | 39 | 42 |
| Ours | **25** | **25** | **26** | **28** | **28** | **30** | **30** | **31** | **32** | **34** |

Table 5: Ablation Study on different components and their impact on AIF. The ExPI dataset and Cross-Trial (CT) protocol were used.

| Method | AIF (↓) |
|---|---|
| No srRPE Query | 0.01648 |
| No srRPE Key | 0.01576 |
| No srRPE Value | 0.01639 |
| No TST | 0.02085 |
| No srRPE | 0.01639 |
| Ours | **0.01521** |

**Spatial Relation-aware Relative Positional Encoding**   In Table 6, we focus on evaluating the components under the SR-MSA block and srRPE on the ExPI dataset using the CT protocol. It can be seen that the AME increases when there is no srRPE Query, srRPE Key or srRPE Value used in the SR-MSA block. We also replaced our proposed srRPE with an MLP Positional Encoding block which resulted in a significant increase in AME. By encoding the proximal nature of the frames of interaction, we are able to generate high quality synthesized motion.

Table 6: Ablation Study on different components under the SR-MSA block. The ExPI dataset and Cross-Trial (CT) protocol were used.

| | AME (averaged across actions) (↓) | | | | | | | | | |
|---|---|---|---|---|---|---|---|---|---|---|
| Time (sec) | 0.2 | 0.6 | 1.0 | 1.4 | 2.0 | 2.4 | 2.8 | 3.2 | 3.6 | 4.0 |
| No srRPE Query | 27 | 27 | 28 | 29 | 29 | 31 | 32 | 33 | 36 | 37 |
| No srRPE Key | 26 | 26 | 27 | **28** | 29 | 31 | 31 | 32 | 37 | 37 |
| No srRPE Value | 27 | 27 | 28 | **28** | 29 | 31 | 32 | 33 | 36 | 37 |
| MLP Positional Encoding | 35 | 35 | 36 | 38 | 39 | 38 | 39 | 39 | 40 | 43 |
| Ours | **25** | **25** | **26** | **28** | **28** | **30** | **30** | **31** | **32** | **34** |

We further demonstrate the importance of the proposed relative position encoding by visualizing the effective receptive field using a heatmap (Figure 1, middle row) and line graph (Figure 1, bottom row) on the cartwheel motion (top row) from the ExPI dataset. , and the representative key poses (top row) are rendered using the corresponding colours. Our TopoFormer is able to learn the topological and proximal relationship across the spatio-temporal domain of the interaction.

## 5   CONCLUSION

In this paper, we propose a new Transformer, called 'TopoFormer', for reactive motion prediction in Two-person close interactions. The proposed Topology-Aware Spatio-Temporal Embedding and Spatial Relation-aware Multi-Headed Self Attention (SR-MSA) effectively earn the topological and proximal relationship across the spatio-temporal domain of the interaction. Experimental results highlight the effectiveness of our method which achieved SOTA performance across different datasets. In the future, we will explore applying our methods to other related tasks such as interaction prediction.

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

## A    APPENDIX - SUPPLEMENTARY MATERIALS

### A.1    DISCRETISATION AND ANALYTICAL SOLUTION FOR GLI CALCULATION

As stated in Section 3.1.1, the GLI of two curves can be computed using Eq. 1 (in the paper). However, it is computationally costly to compute the GLI for long curves since there is a double integral in the equation. On the other hand, an analytical solution Levitt (1983) can be used when chained segments approximate the curves. Here, we first discretise the curves by representing each character using two sets of chains to capture the limb-level and body-level interactions as explained in Section 3.1.2 in the paper.

Given two chains, $S1$ and $S2$, with $m$ and $n$ line segments on the chains, respectively. The GLI can be calculated by

$$GLI(S_1, S_2) = \sum_{i=1}^{m} \sum_{j=1}^{n} T_{i,j}$$

where $T_{i,j}$ is the writhe of segment $i$ and segment $j$.

The analytical solution Levitt (1983) can be used for calculating $T_{i,j}$. Specifically, given line segments $i$ (with end points $a$ and $b$) and $j$ (with endpoints $c$ and $d$), we can define 6 vectors as $r_{ab}$ $(a \rightarrow b)$, $r_{ac}$ $(a \rightarrow c)$, $r_{ad}$ $(a \rightarrow d)$, $r_{bc}$ $(b \rightarrow c)$, $r_{bd}$ $(b \rightarrow d)$, $r_{cd}$ $(c \rightarrow d)$. The vectors will be used for calculating the normal vectors of the tetrahedron created using those 4 endpoints:

$$n_a = \frac{r_{ac} \times r_{ad}}{|r_{ac} \times r_{ad}|}, n_b = \frac{r_{ad} \times r_{bd}}{|r_{ad} \times r_{bd}|},$$
$$n_c = \frac{r_{bd} \times r_{bc}}{|r_{bd} \times r_{bc}|}, n_d = \frac{r_{bc} \times r_{ac}}{|r_{bc} \times r_{ac}|}.$$

Finally, $T_{i,j}$ is calculated by

$$T_{i,j} = \arcsin(n_a n_b) + \arcsin(n_b n_c) + \arcsin(n_c n_d)$$
$$+ \arcsin(n_d n_a).$$

### A.2    EXPERIMENTAL RESULTS

#### A.2.1    VISUALIZATION OF THE LATENT SPACE

We further visualize the latent representation learned using our proposed TopoFormer Encoder in Figure 5. It can be seen that each motion is represented in a longer clustered trajectory where each point corresponds to the encoding of the poses in a given sequence. This aligns well with the rationale behind the design as now the motions are having smoother transitions in the topology-informed interaction space.

#### A.2.2    QUALITATIVE EVALUATION

The quality of the generated interactions can also be evaluated qualitatively in Figure 6 to 11. Readers are referred to the accompanying video demo for the predicted reactive motion quality and comparison with STOA.

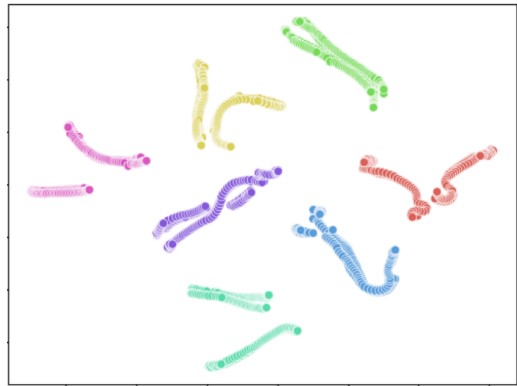

Figure 5: t-SNE representation of the topological aware latent space from our TopoFormer Encoder for ExPI (CT) experiment. Each color represents an input motion of the same class interaction.

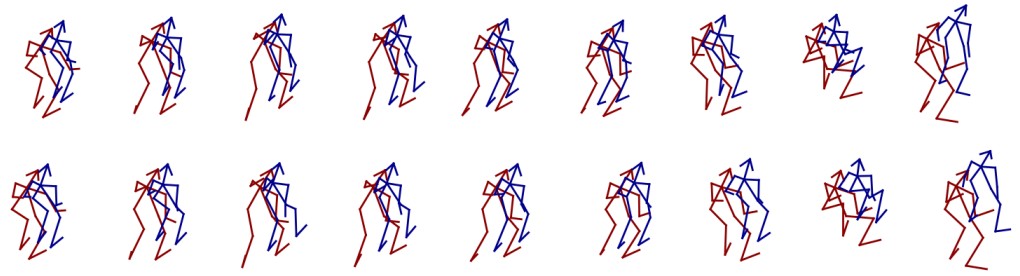

Figure 6: Interaction Motion for ExPI (Cross-Trial) a-frame. Top row is the ground truth. Bottom row is our synthesized results.

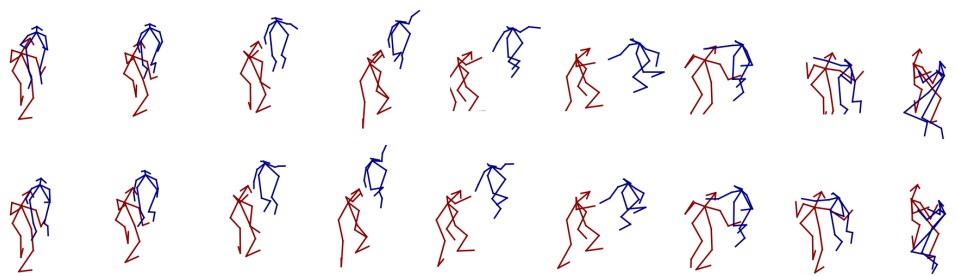

Figure 7: Interaction Motion for ExPI (Cross-Trial) aroundtheback. Top row is the ground truth. Bottom row is our synthesized results.

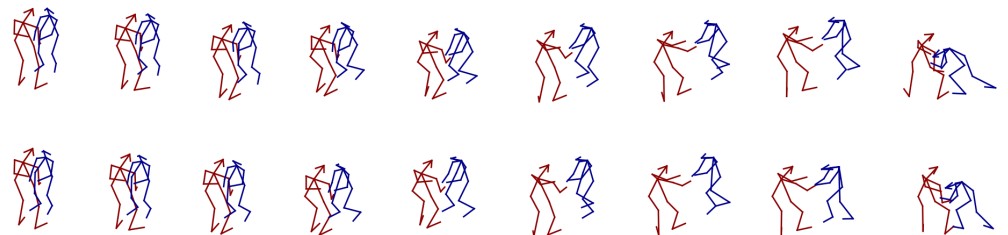

Figure 8: Interaction Motion for ExPI (Cross-Trial) cartwheel. Top row is the ground truth. Bottom row is our synthesized results.

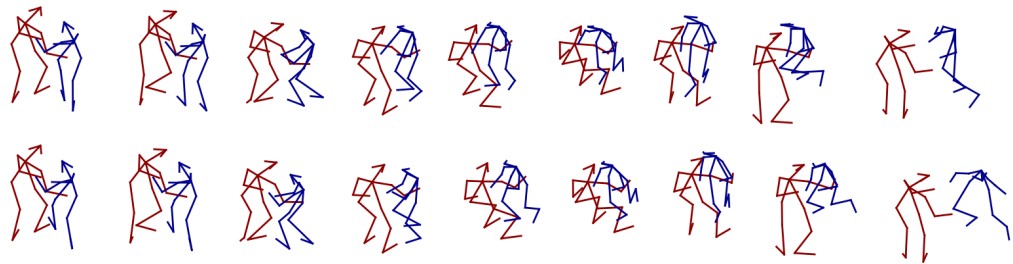

Figure 9: Interaction Motion for ExPI (Cross-Trial) toss-out. Top row is the ground truth. Bottom row is our synthesized results.

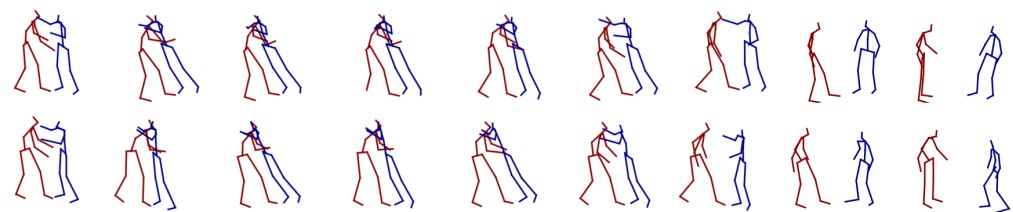

Figure 10: Interaction Motion for CHI3D (Cross-Trial) hug. Top row is the ground truth. Bottom row is our synthesized results.

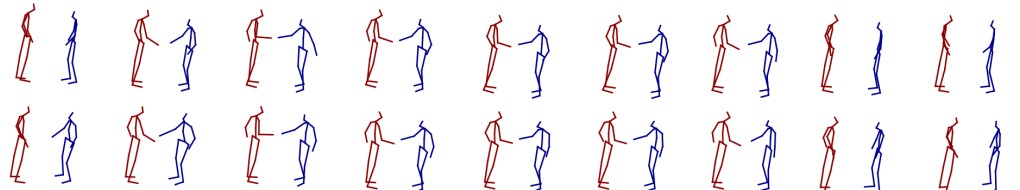

Figure 11: Interaction Motion for CHI3D (Cross-Trial) handshake. Top row is the ground truth. Bottom row is our synthesized results.

