# OpenReview forum: "TopoFormer: Topology-aware Transformer for Reactive Motion Prediction in Close Interactions"
_ICLR.cc/2024/Conference — ICLR 2024 Conference Withdrawn Submission_

### Official Review · Reviewer_MoxH · 2023-10-29

**Soundness:** 2 fair
**Presentation:** 2 fair
**Contribution:** 2 fair
**Rating:** 5
**Confidence:** 3

**Summary:**

This paper introduces a novel Transformer model called TOPOFORMER for predicting a character's reactive actions during intimate interactions. The authors propose a topology-aware spatio-temporal embedding and a spatial relations-aware multi-head self-attention mechanism to facilitate the learning of latent representations for intimate interactions. Spatial relationships can be represented more efficiently by using a set of joint chains to represent body parts instead of the graph-based structure commonly used in recent work.The experimental results show that TOPOFORMER achieves state-of-the-art performance on different datasets. The authors also mentioned that they will further investigate applying the method to other related tasks such as interaction prediction.

**Strengths:**

1. A new Transformer model is proposed, which takes actions and interactive category tags of another character as input. Learn potential representations of close interactions through structures that include Topology-Aware Spatio-Temporal  embedments and Spatial Relation-aware Multi-Headed Self-Attention .
2. Represent spatial relationships more effectively by using a set of joint chains to represent body parts, rather than the graph-based structure commonly used in recent studies. By using the Gauss Linking Integral  to indicate proximity to body parts, minimizing the variation of the paired GLIs on successive frames, mutual penetration of body parts can be avoided.
3. A continuous parameterized topological feature is proposed to describe interactive semantics.

**Weaknesses:**

1. The figure is rough and the resolution of the image is relatively low.The meaning of some markers and curves in the figures is not clear and there are some mistakes in figures,e.g. in figure 2，SRA-MSA block is inconsistent with the SR-MSA block in the following section.
2. The innovation point is insufficient and the improvement of the main network structure is relatively small.
3. The comparison experiment is not enough to prove the effect of the algorithm.The models involved in the experiment are few and unconvincing.The dataset used in the experiment is not very general in this field.
4. The newly proposed evaluation metric is specifically aimed at GLI, and this paper mainly optimizes this problem, which is not generalized compared with other models.

**Questions:**

1. For the content in weakness 1，I would like to know the meaning of the ordinate of the line chart at the bottom of Figure 1.And the numbers in the picture are deformed.
2. For the content in weakness 3,In the papers of several models compared by the author, no relevant experiments on AME and AIF evaluation indexes were found. Why does the author say that this model is sota？

---

### Official Review · Reviewer_Puux · 2023-10-31

**Soundness:** 2 fair
**Presentation:** 2 fair
**Contribution:** 2 fair
**Rating:** 3
**Confidence:** 4

**Summary:**

For the task of multi-person interactive motion prediction, the author proposes a Transformer structure based on existing methods, which mainly includes Spatial Relation-aware Multi-Headed Self Attention (SR-MSA) and a topological feature with continuous parameterization. Some experiments have been conducted to test the proposed TopoFormer structure.

**Strengths:**

1. Theoretical analysis and experimental verification of the important role of topological relations in interactive motion are carried out, and the validity of the proposed Topology-Aware Spatio-Temporal block is verified both qualitatively and quantitatively.
2. Ablation experiments verify that the combination of the proposed modules can obtain fine results on the validation dataset.

**Weaknesses:**

1. The core contribution of the paper is not presented intuitively and completely, which makes the reader unable to accurately grasp the core contribution.
2. The author introduced the core modules of the paper in sections 3.2, 3.3 and 3.4, but it lacking specific illustrations and formula descriptions.
3. The innovation of this work is not good, where the Transformer structure in this work is based on the existing methods.
4. The performance is not outstanding, where the improvements are not significant.

**Questions:**

1.	In the experiments that compared with the SOTA algorithms, evaluation indexes such as accuracy which were not designed in this paper are used. The author should explain why the evaluation indexes used in this paper are chosen instead of other evaluation indexes.
2.	For this interactive task, apart from the quantitative comparison of results, why not consider the visual comparison of the model results? This comparison should be able to more intuitively represent the effectiveness of the new topology proposed in this paper.
3.	In the introduction part, the authors explain the effectiveness of the proposed topology structure compared with the existing topology features. However, the ablation experiment can only prove the effectiveness of the topology structure, but cannot quantitatively prove the superiority of the proposed topology structure compared with the existing topology features.

**Details Of Ethics Concerns:**

I do not have the Ethics Concerns

---

### Official Review · Reviewer_faxm · 2023-11-01

**Soundness:** 2 fair
**Presentation:** 1 poor
**Contribution:** 2 fair
**Rating:** 3
**Confidence:** 4

**Summary:**

This paper proposed a transformer-based framework for predicting the reactive motion of one of the characters in a Two-person close
interaction by giving the motion of the other character and the interaction class label as input. The proposed method represents
the body parts using a set of articulated chains topology-based representations, Gauss Linking Integral. Based on the topology-based representation, the authors further proposed a Topology-Aware Spatio-Temporal Embedding and Spatial Relation-aware Multi-Headed Self Attention (SR-MSA) to learn of the latent representation of close interactions. Experimental results highlight the effectiveness of the proposed method in Aligned Mean Error (AME) and a newly proposed metric Average Interpenetration per Frame (AIF) across different datasets.

**Strengths:**

1. This paper proposed to represent the body parts using a set of articulated chains instead of the commonly
used graph-based structure in recent works to capture the semantic information in interactive motions.
2. Experimental results showed the sota performance compared to the existing method across different datasets.

**Weaknesses:**

1. This paper uses a set of articulated chain topology-based representations, Gauss Linking Integral, to represent the body parts. However, there lack of clear mathematical explanation of this representation to capture the semantic information, compared to the commonly used joints-based and mesh-based representation. More in-depth analysis and comparisons should be added to support the argument for the task.
2. The proposed module Topology-Aware Spatio-Temporal Embedding applies an MLP-based structure, of which the technical contribution is minor. The benefits of the proposed two modules Topology-Aware Spatio-Temporal Embedding and Spatial Relation-aware Multi-Headed Self Attention are unclear. What are the roles of each module? The authors claim both can capture the semantic information, then regarding the penetration, which design contributes to it?
3. Regarding the Ablation in experiments: Tables 4 and 5 show the "drop-one-out" results, I would also like to see "incremental" results by adding each component one by one.
4. More qualitative comparisons to existing methods should be shown to support the argument, instead of only comparing to GT.

**Questions:**

See the above weaknesses.

---

### Official Review · Reviewer_zYBk · 2023-11-04

**Soundness:** 3 good
**Presentation:** 2 fair
**Contribution:** 3 good
**Rating:** 5
**Confidence:** 4

**Summary:**

This paper presents a Transformer, TopoFormer, for predicting the reactive motion of one of the characters in a two-person close interaction. It consists of a Topology-Aware Spatio-Temporal (TST) embedding and Spatial Relation-aware Multi-Headed Self-Attention (SR-MSA) Instead of the commonly used graph-based structure, it uses a set of articulated chains to represent the body parts. In this way, the spatial relations can be represented using a topology-based representation, i.e., Gauss Linking Integral (GLI). TopoFormer was evaluated on two public datasets, i.e., ExPI and CHI3D, and achieved good experimental results.

**Strengths:**

- I like the idea of encoding the pairwise GLIs of closely interacting body parts to prevent implausible motions.
- Good ablation study and experimental results are reported.
- The literature review covers representative works of each related domain.

**Weaknesses:**

- The methodology is expected to be general, however, the authors refer to the topological features as a fixed number (Section 3.1, 3.2). I understand that this approach is evaluated on the two datasets. It would be better to denote them with symbols and move the real numbers to  Section 4. The same issue happens with the numbers of serial chains and network layers.
- It is still unclear why GLI is a better way to represent human motion although it is “inspired by the success of using GLI as soft constraints in an optimization-based approach for generating collision-free close interactions”. There is no theoretical analysis or mathematical proof.
- Please be serious about using math. There are no explanations of the meaning of $S, A$ (Section 3.1) or $T, D$.
- It is unclear how TPE and FPE work. More details would be better.
- The first reference to Figure 1 is in Section 4.5. It does not make sense to put it as the first figure. It is unclear what the colors on top stand for. Which ones are highlights?
- Figure 4 is confusing. As is shown in the ground truth, the starting of the tumbling motion of the cartwheel happens at 6/7 of the motion. MLP takes it at the first 10% while the proposed sr-RPE highlights 40% - 60%. Both of them failed to capture the correct position.
- It will be good to know the impact of TPE and FPE individually.
- Typos:
    - I guess you mean “learn” when saying “effectively ‘earn’ the topological and proximal relationship”.
    - An extra dot in the last sentence of Section 3.3.

**Questions:**

Since you try to learn a non-Euclidean (topological) feature, why do you still use an Euclidean error (MPJPE) to optimize the loss?

**Details Of Ethics Concerns:**

Please see the questions and weaknesses.

---

### Meta-Review · Area_Chair_54jW · 2023-12-09

**Metareview:**

No rebuttal is provided. The AC agrees with the reviewer to reject the paper.

**Justification For Why Not Higher Score:**

No rebuttal is provided. The AC agrees with the reviewer to reject the paper.

**Justification For Why Not Lower Score:**

N/A

---

### Decision · Program_Chairs · 2024-01-16

Reject